# RuBQ 2.0: An Innovated Russian Question Answering Dataset

Ivan Rybin[1][*], Vladislav Korablinov[1,2], Pavel Efimov[1], and Pavel Braslavski[3,4][0000−0002−6964−458X]

[1] ITMO University, Saint Petersburg, Russia
{isrybin,vladislav.korablinov,pavel.vl.efimov}@gmail.com
[2] Yandex, Saint Petersburg, Russia
[3] Ural Federal University, Yekaterinburg, Russia
[4] HSE University, Moscow, Russia
pbras@yandex.ru

**Abstract.** The paper describes the second version of RuBQ, a Russian dataset for knowledge base question answering (KBQA) over Wikidata. Whereas the first version builds on Q&A pairs harvested online, the extension is based on questions obtained through search engine query suggestion services. The questions underwent crowdsourced and in-house annotation in a quite different fashion compared to the first edition. The dataset doubled in size: RuBQ 2.0 contains 2,910 questions along with the answers and SPARQL queries. The dataset also incorporates answer-bearing paragraphs from Wikipedia for the majority of questions. The dataset is suitable for the evaluation of KBQA, machine reading comprehension (MRC), hybrid questions answering, as well as semantic parsing. We provide the analysis of the dataset and report several KBQA and MRC baseline results. The dataset is freely available under the CC-BY-4.0 license.

**Keywords:** Knowledge base question answering · Multilingual question answering · Machine reading comprehension · Evaluation resources · Russian language resources
**Resource location:** https://doi.org/10.5281/zenodo.4345696
**Project page:** https://github.com/vladislavneon/RuBQ

## 1  Introduction

Question Answering (QA) is an important scientific and applied problem that aims at building a system that can answer questions in a natural language. Two main directions within QA are *Open-Domain Question Answering (ODQA)* and *Knowledge Base Question Answering (KBQA)* (also referred to as *Knowledge Graph Question Answering, KGQA*). ODQA searches for the answer in a large collection of text documents; the process is often divided into two stages: 1) retrieval of documents/paragraphs potentially containing the answer and 2) spotting

---

[*] Work done as an intern at JetBrains Research.

an answer span within a given document/paragraph (referred to as *machine reading comprehension, MRC*). In contrast, KBQA searches an answer in a knowledge base that is commonly structured as a collection of *(subject, predicate, object)* triples, e.g. *(Yuri Gagarin, occupation, astronaut)*. The KBQA task can be formulated as a translation from a natural language question into a formal semantic representation, e.g. a SPARQL query. In many real-life applications, like in *Jeopardy!* winning IBM Watson [11] and major search engines, hybrid QA systems are employed – they rely on both text document collections and structured knowledge bases.

As is typical for all machine learning applications, freely available annotated data for model training and testing is crucial for measurable progress in solving QA tasks. Since the inception of the SQuAD dataset [18], we have seen an avalanche of the available question answering datasets; the most recent trend is non-English and multilingual MRC datasets [16, 15, 4]. Available KBQA datasets are much scarcer; there are very few non-English datasets among them (see Section 2 for details). Russian is among top-10 languages by its L1 and L2 speakers;[5] it has a Cyrillic script and a number of grammar features that make it quite different from e.g. English and Chinese – the languages most frequently used in NLP and Semantic Web research.

In this paper, we describe an expansion of **RuBQ** (pronounced ['rubik]) – **Ru**ssian Knowledge **B**ase **Q**uestions, the first Russian KBQA dataset [13]. The first version of the dataset is based on the pairs of questions and answers from online quizzes and contains 1,500 questions along with Wikidata answers, corresponding SPARQL queries, and English machine-translated questions. The dataset is accompanied with a Wikidata snapshot `RuWikidata8M` containing about 212M triples that include 8.1M unique items with Russian labels.

To expand the dataset, we obtained questions from Yandex and Google query suggestion services. A potential advantage of such questions from a search log is that they reflect realistic users information needs and their wordings. A possible disadvantage of query suggestion services as a source of questions is that we have very little control over the questions' selection criteria, as well as a limited number of returned suggestions for each input. In addition, the services rank the returned queries by their popularity, which may result in shorter, simpler, and less varied questions. As a seed, we used a set of manually crafted question prefixes corresponding to the most popular Wikidata properties, as well as properties with numeric values. To annotate the questions, we employed a pipeline quite different from the version one. We used crowdsourcing to find the answers to the questions and generated SPARQL queries based on automatically extracted question entities and the corresponding properties. We retained those items that produced correct answers. Using this routine, we were able to almost double the dataset: the extended version reached 2,910 questions.

In addition, using automatic methods and subsequent crowdsourced verification, we coupled most of the questions in the dataset with Wikipedia paragraphs

---

[5] `https://en.wikipedia.org/wiki/List_of_languages_by_total_number_of_speakers`

containing the answer. Thus, the dataset can be used for testing MRC models, as well as for research on hybrid question answering. We provide several baselines both for KBQA and MRC that demonstrate that there is an ample room for improving QA methods. In the reminder of the paper we will refer to the add-on to the initial dataset as *Query Suggestion Questions* (QSQ), and the merging of RuBQ 1.0 and QSQ as RuBQ 2.0. Taking into account RuBQ's modest size, we propose to use the dataset primarily for testing rule-based systems, models based on few/zero-shot and transfer learning, as well as models trained on automatically generated examples, similarly to recent MRC datasets [1, 15]. RuBQ 2.0 is freely available under the CC-BY-4.0 license in JSON format.[6]

## 2    Related Work

*KBQA datasets.* In contrast to the field of MRC, available KBQA resources for training and evaluation are fewer in numbers. For a comprehensive survey of KBQA datasets, including the first edition of RuBQ, see our paper [13]. As the survey shows, there are less than 20 publicly available KBQA datasets to date; the majority of them are in English. Few exceptions are Chinese MSParS [7], multilingual (machine-translated to a large extent) QALD [21], and the newly introduced Russian RuBQ. The most recent dataset, Multilingual Knowledge Questions and Answers (MKQA), was published in the summer of 2020 [16]. The dataset contains human translations of 10,000 English questions from the Natural Questions (NQ) dataset [14], whose entries originate from Google search log, into 25 languages, including Russian. MKQA refrains from using the existing NQ answers and obtains the answers anew for each language variant via crowdsourcing. The answers are divided into several types (in descending order by frequency): entity, long answer, unanswerable, date, number, number with unit, short phrase, as well as yes/no. Entity answers (4,221 in the English subset, for other languages the number may slightly differ) are linked to Wikidata. However, the dataset does not contain annotations of the question entities nor corresponding formal representations such as SPARQL queries.

*WebQuestions.* Our approach uses search engine query suggestion services to obtain new questions and is similar to the approach behind the WebQuestions dataset containing 5.8K questions [2]. Later, 81% of WebQuestions were provided with SPARQL queries and formed the WebQuestionsSP dataset [22]. In contrast to WebQuestions that started with a single question and iteratively expanded the pool of questions, we started with a manually crafted list of question prefixes presumably corresponding to the most frequent Wikidata properties and properties with literal values.

*MRC datasets.* Since the advent of SQuAD [18], a large number of MRC datasets have been published. In the context of our study, the most relevant datasets are those built in a semi-automatic manner using existing Q&A pairs. TriviaQA [12]

---

[6] https://github.com/vladislavneon/RuBQ

builds on a collection of 95K trivia and quiz question-answers pairs collected on the Web. The dataset is enriched with documents from Web search results and Wikipedia articles for entities identified in the questions, 650K documetens in total. These evidence documents are not guaranteed to be sufficient to answer the questions. A manual evaluation of about 1K questions showed that the retrieved evidence indeed contains correct answers in 79.7% and 75.4% cases for Wikipedia and Web documents, respectively. SearchQA dataset [9] exploits a similar approach: it starts from a collection of 140K question-answer pairs from *Jeopardy!* archive and is augmented with Google search snippets containing the answer. Each SearchQA Q&A pair is aligned with about 50 such snippets on average. In contrast to these datasets, we conducted an exhaustive crowdsourced annotation of the evidence paragraphs.

*Russian MRC datasets.* Built in 2017 with 50K question-paragraph-answer triples and using SQuAD as a reference, SberQUAD is the largest Russian MRC dataset. A post-hoc analysis revealed that the dataset is quite noisy [10]. Multilingual XQuAD [1] and TyDi QA [4] datasets contain around 1K and 7K Russian items, respectively; summary statistics and comparison of these three resources can be found in our recent paper [10].

## 3   Data Acquisition and Annotation

*Questions.* To obtain new questions, we used query suggestion services by Google and Yandex search engines. Firstly, we consulted a list of the most popular Wikidata properties,[7] removed cross-linking properties such as *PubMed ID* and *DOI*, and manually crafted question prefixes for the remaining top-200 properties, 2,077 in total. Examples include: *Under what license is X distributed...* or *What instrument did X play...*[8] Often, the prefixes for the same property are almost identical – they differ in the present/past tense, singular/plural or reflexive/non-reflexive verb forms. Note that these prefixes can be ambiguous and do not uniquely define the property. In addition, we manually compiled 546 question prefixes that imply numerical and date answers, for example: *What is melting point...* and *When was X invented...* Search engine suggestion services return up to 10 items for each input. Sometimes the beginnings of returned items do not match the initial prefixes, which is likely due to semantic rather than lexical matching methods. Given a limited number of returned queries, the simultaneous use of two services allowed to slightly increase the variety of questions. In total, we collected 18,262 queries for prefixes corresponding to popular properties, and 3,700 queries for prefixes with expected numerical answers. These queries have been filtered semi-automatically: we removed the duplicates, queries without question words, and questions that cannot be answered using solely a knowledge

---

[7] https://www.wikidata.org/wiki/Wikidata:Database_reports/List_of_properties/all

[8] Original Russian prefixes don't include $X$s – according to Russian word order the subject comes next.

base, e.g. *What holiday is it today?* After such cleaning, 4,069 unique questions remained, corresponding to 146 properties from the top-200 Wikidata properties, and 1,685 questions with expected numerical answers. We refer to these question samples as *TopProperties* and *LiteralValues*, respectively.

*Answers.* To obtain the answers to the questions, we posted a project on the Toloka crowdsourcing platform.[9] The crowd workers' task was to find one or several answers to a question on the Web or indicate that they cannot find any. Crowd workers entered the answers as strings; in case of multiple answers they separated them with a comma. The interface contained a link to the Google search results for the question, but the workers were instructed that they are free to modify the search query and use other information sources on the Web. Each question was shown to three workers. For 300 questions out of 4,069 (7.4%) *TopProperties* questions, the majority of workers (two or three) failed to find an answer. For the remaining 3,769, in 1,956 (48.1%) cases all three workers provided identical answers, in 1,278 (31.4%) cases two answers matched. Thus, we obtained the answers for 3,234 questions. In addition, we asked the crowd workers to indicate whether the answer was found as a Google instant answer (also called 'features snippet'). The share of such answers was 65.3%, i.e. the search engine answers about 2/3 of the questions using question answering capabilities. We obtained reliable answers for 1,537 out of 1,685 *LiteralValues* questions: in the case of 1,153 (68.4%) questions all workers were unanimous; on 384 (22.8%) questions two out of three agreed. The share of instant answers in the Google results for this group was higher – 87.8%, the share of unanswered questions was 2.7%.

*Entity linking and SPARQL queries.* Firstly, we applied an IR-based entity linker developed for RuBQ 1.0. For each input string, the linker generates several phrase and free-form queries to a search index of all Russian labels and aliases from Wikidata (about 5.4M items). Returned Wikidata entity candidates are ranked based on a combination of confidence value from the search engine and the popularity of the corresponding Wikipedia pages. The linker proved to be efficient and of high quality, details can be found in RuBQ 1.0 paper [13]. For the current step we retained top-5 candidates for each question and answer. Secondly, using the question's anticipated property (let us remind you that questions' prefixes are made up to reflect specific properties), we checked whether a triple *(question_entity, property, answer_entity)* with any combination of candidate question/answer entities was present in Wikidata. This fully automatic procedure resulted in 746 questions linked to Wikidata (407 in *TopProperties* and 339 in *LiteralValues*). This approach is quite different from the one of RuBQ 1.0, where entity linking in answers was performed by crowd workers. To increase the number of annotated questions and answers, we manually corrected some questions' properties and entities. Note that prefixes sent to the query suggestion service do not guarantee that the returned question expresses the intended property.

---

[9] https://toloka.ai/

This selective in-house annotation allowed us to reach the target number of 1,200 annotated questions.

**Table 1.** Dataset statistics.

|                                            | RuBQ 1.0 | QSQ   | RuBQ 2.0 |
|--------------------------------------------|----------|-------|----------|
| Questions                                  | 1,500    | 1,410 | 2,910    |
| KB-answerable questions                    | 1,200    | 1,200 | 2,400    |
| KB-unanswerable questions                  | 300      | 210   | 510      |
| Avg. question length (words)               | 7.9      | 5.4   | 6.7      |
| Simple questions (*1-hop* w/o aggregation) | 921      | 911   | 1,832    |
| Questions with multiple answers            | 131      | 244   | 375      |
| Questions with literals as answers         | 46       | 600   | 646      |
| Unique properties                          | 242      | 139   | 294      |
| Unique entities in questions               | 1,218    | 1,126 | 2,114    |
| Unique entities in answers                 | 1,250    | 1,983 | 3,166    |

*Unanswerable questions.* Since their introduction in SQuAD 2.0 [17], unanswerable questions are featured in many MRC datasets. In the context of an MRC task, such question cannot be answered based on a given paragraph. RuBQ 1.0 and the aforementioned MKQA dataset also contain unanswerable questions. In case of RuBQ, unanswerable questions cannot be answered using the provided Wikidata snapshot. In case of MKQA, this category encompasses ill-formed questions and questions for which no clear answer can be found. We also enriched QSQ with questions, for which the majority of crowd workers agreed on the answer, but the question could not be answered with the current state of Wikidata graph. While many unanswerable quiz questions from the first version of RuBQ can be hardly expressed with Wikidata properties (e.g. *How many noses do snails have?*)[10], unanswerable QSQs are quite common in their form and cannot be answered rather due to KB incompleteness, e.g. *Who is the inventor of streptomycin?*

*Dataset statistics.* As can be seen from the Table 1, using the process described above, we were able to almost double the size of the dataset: the number of questions that can be answered using the current version of Wikidata has doubled, while we added slightly fewer unanswerable questions compared to the first version (210 vs. 300). Questions from search engine suggestion services are significantly shorter than quiz questions. Due to the procedure of matching the answers to the Wikidata entities in the first version of the dataset, there were very few questions with numerical and date answers. When working on the extension, we specifically addressed this problem – half of the added answerable questions have a literal answer. Following RuBQ 1.0, we annotated the questions based on the structure of corresponding SPARQL-queries. As in the previous version of the dataset, the majority of questions (911) are *simple*, i.e. they are one-hop questions without

---

[10] The answer is four; Google returns an instant answer with a supporting piece of text.

**Table 2.** Sample RuBQ 2.0 entries. Questions are originally in Russian; original aliases in the bottom example are partly in Russian, partly in English; not all fields of the dataset are shown.

| Question | How many wives did Henry VIII have? |
|---|---|
| Answer | 6 |
| Answer aliases | six |
| SPARQL query | `SELECT (COUNT(?x) as ?answer)` 
 `WHERE {` 
 `   wd:Q38370 wdt:P26 ?x.` 
 `   }` |
| Tags | count, 1-hop |
| Paragraphs with answers | 51089, 51086 |
| Related paragraphs | 51086, 51087, 51088, 51089... |
| RuBQ version | 1 |
| Question | Where is Kutuzov buried? |
| Answer | Q656 (Saint Petersburg) |
| Answer aliases | Leningrad, Petersburg, St. Petersburg, Petrograd, Sankt-Peterburg |
| SPARQL query | `SELECT ?answer` 
 `WHERE {` 
 `   wd:Q185801 wdt:P119 ?answer` 
 `      }` |
| Tags | 1-hop |
| Paragraphs with answers | 416 |
| Related paragraphs | 416, 417, 418, 419, 420... |
| RuBQ version | 2 |

aggregation.[11] While the share of simple questions is about the same as in the first version, the number of questions with list answers increased significantly. Even though the number of questions doubled, this led to only a moderate increase in unique properties in the dataset – from 242 in the first version to 294 in the second. At the same time, the number of unique entities in questions has almost doubled; the increase in the number of unique answer entities was even more significant. We divided QSQ into dev (280) and test (1,130) subsets maintaining similar ratios of questions types in the subsets. Thus, RuBQ 2.0 contains 2,330 and 580 questions in test and dev sets, respectively. The dataset is available in JSON format; sample entries are shown in Table 2.

## 4   Adding Machine Reading Capabilities

To enrich the dataset with machine reading capabilities, we collected all Russian Wikipedia articles corresponding to question and answer entities and split them into paragraphs. Next, we ranked the paragraphs by decreasing the likelihood of

---

[11] Our approach resulted in a slightly lower share of simple questions in QSQ compared to WebQuestions: 76% vs. 85%. It can be attributed to the source of questions and the collection process, as well as to differences in Freebase vs. Wikidata structure.

containing the answer to the question. To do this, for each question and answer entity, we compiled a list of its name variants. The list included Russian and English labels and aliases ("alternate names") from Wikidata, as well as the Wikipedia anchor texts pointing to the entity page. In most cases, the anchor text was an inflectional variant of the entity name (e.g. *Alexandra Pushkina $_{ru}$* – genitive case of *Alexander Pushkin*), short form (*Pushkin* for *Alexander Pushkin*), or a cross-POS relationship (for example, *Russian $_{adj}$* pointing to *Russia $_{noun}$*). Thus, we obtained a representation for the question and answer as the lemmatized[12] source text with stopwords removed and entity name synonyms added.

The paragraphs were ranked in descending order of word occurrences from the answer representation; within equal number of occurrences the paragraphs were ranked according to the word occurrences from the question. For unanswerable questions with no entities linked, we ranked paragraphs from the top-5 Wikipedia search API results[13] using the whole question as a query. Further, the top-4 paragraphs containing at least an answer evidence were annotated on the Toloka crowdsourcing platform. The paragraphs were presented to the crowd workers along with the original question. The workers' task was to mark all paragraphs that provide enough information to answer the question. Paragraphs were annotated for 2,754 questions out of total 2,910: there were no answer hits in paragraphs corresponding to 156 questions (in these cases the answer was often present not in the article body, but in its infobox). For 2,131 questions, crowd workers marked at least one paragraph as providing the answer; on average, there are 1.35 supporting paragraphs per question; 623 questions are not provided with an answer-bearing paragraph. Note that 186 (out of 510) KB-unanswerable questions are provided with answer-bearing paragraphs, which can be seen as a potential for hybrid QA.

To be able to use the dataset not only to assess machine reading comprehension, but also paragraph retrieval, we added related paragraphs for each answer. For each question, we added up to 25 paragraphs from top-5 Wikipedia search results using question as a query, ranked by the occurrence of words from the answer and question representations in a similar way to how we ranked paragraphs to send them to crowdsourcing. In total, we provide 56,952 unique Wikipedia paragraphs as a part of the dataset; the average length of a paragraph is 62 tokens.

The data preparation process described above has several advantages. Firstly, the annotation process is simpler and faster compared to SQuAD and similar datasets: crowd workers do not need to select a span within a paragraph – only mark a paragraph as sufficient to answer the question. Secondly, the questions are generated independently from the paragraph, which leads to a more natural and a more realistic task. When questions are generated by crowd workers based on the given paragraphs like in SQuAD or SberQuAD, they tend to be lexically and structurally similar to sentences containing answers and easier to answer as such, see discussion in the paper describing Natural Questions dataset [14]. In

---

[12] We used *mystem* lemmatizer, see `https://yandex.ru/dev/mystem/` (in Russian).
[13] `https://ru.wikipedia.org/w/api.php`

contrast to MRC datasets produced fully automatically based on Q&A pairs and search, relevant paragraphs in RuBQ 2.0 are verified manually.

## 5  Baselines

### 5.1  KBQA Baselines

We evaluated four systems that return responses from Wikidata and implement different approaches to KBQA.

*DeepPavlov* is an open library featuring models for variety of NLP tasks, including Russian language models [3]. We tested the previous DeepPavlov KBQA system that coped only with simple questions, on RuBQ 1.0 [13]. An improved KBQA model was released by DeepPavlov in summer of 2020.[14] The new release has several components and addresses not only simple questions, but also ranking and counting questions, as well as questions requiring reasoning on numerical values and dates. According to its description, the system sequentially performs query type prediction, entity extraction and linking, relation extraction and path ranking, and finally issues an online SPARQL query to Wikidata.

*QAnswer* is a rule-based KBQA system that answers questions in several languages using Wikidata [6]. QAnswer returns a (possibly empty) ranked list of Wikidata item IDs along with a corresponding SPARQL query. Since recently, QAnswer accepts Russian questions. We obtained QAnswer's results by sending either original Russian or English machine-translated questions to its API.[15]

*Simple baseline.* We also provide our own simple rule-based baseline addressing simple questions.[16] The method consists of three components: 1) entity linker based on syntax parsing and a search index, 2) rule-based relation extraction, and 3) SPARQL query generator. The question is parsed with the DeepPavlov parser;[17] interrogative word/phrase is identified using a dictionary, and remaining subtrees from the root are candidate entity mentions. These candidates are mapped to Wikidata entities using a search index. Relation extraction method is quite straightforward: we compiled a list of 100 most frequent Wikidata properties and manually generated regular expressions for them.[18] A question can match regular expressions corresponding to different properties, all of them are added as candidates. Based on the obtained lists of candidate entities and properties, 1-hop SPARQL queries are constructed for all entity–relation pairs. Once the query returns a non-empty answer from Wikidata, the process terminates and the answer is returned.

---

[14] http://docs.deeppavlov.ai/en/master/features/models/kbqa.html

[15] https://qanswer-frontend.univ-st-etienne.fr/

[16] https://github.com/vladislavneon/kbqa-tools/rubq-baseline

[17] http://docs.deeppavlov.ai/en/master/features/models/syntaxparser.html

[18] Although these regular expressions and prefixes for collecting QSQs were developed independently and for different sets of properties, this approach can introduce bias in results.

**Table 3.** Baseline results on the tests subsets. DP – DeepPavlov; QA-Ru and QA-En – Russian and English versions of QAnswer, respectively; SimBa – our simple baseline. Detailed tag descriptions can be found in RuBQ 1.0 paper [13].

|  | DP | QA-Ru | QA-En | SimBa |
|---|---|---|---|---|
| **RuBQ 1.0** | | | | |
| **Answerable (960)** | 277 | 222 | 211 | 258 |
| **Unanswerable (240)** | 154 | 9 | 20 | 219 |
| **Total (1,200)** | 431 | 231 | 231 | 477 |
| **QSQ** | | | | |
| **Answerable (960)** | 190 | 295 | 330 | 227 |
| **Unanswerable (170)** | 62 | 15 | 9 | 137 |
| **Total (1,130)** | 252 | 310 | 339 | 364 |
| **RuBQ 2.0** | | | | |
| **Answerable (1,920)** | 467 | 517 | 541 | 485 |
| 1-hop (1,460) | 446 | 449 | 471 | 472 |
| 1-hop + reverse (10) | 0 | 0 | 1 | 0 |
| 1-hop + count (3) | 0 | 1 | 2 | 0 |
| 1-hop + exclusion (17) | 0 | 1 | 1 | 0 |
| multi-constraint (304) | 15 | 60 | 62 | 11 |
| multi-hop (55) | 2 | 6 | 3 | 1 |
| qualifier-constraint (22) | 4 | 0 | 0 | 1 |
| **Unanswerable (410)** | 216 | 24 | 29 | 356 |
| **Total (2,330)** | 683 | 541 | 570 | 841 |

*Evaluation.* We calculated the number of correctly answered questions in respective test sets by each of the system. If QAanswer returned a ranked list of answers, only the top answer was considered. In case of multiple correct answers, a system's answer was counted if it matched any of the reference answers. In case of unanswerable questions, an answer was deemed correct if the systems returned either an empty or "no answer/not found" response. DeepPavlov returns an answer as a string rather than a Wikidata item ID, or *not found*, so we compared its answers with the correct answer's label if the correct answer wss an entity or with its value if the correct answer was a literal.

*Results.* As one can see from the Table 3, the models behave quite differently on answerable questions of the two parts of the dataset. The quality of both versions of QAnswer improves significantly on QSQ compared to RuBQ 1.0, with the improvement of the English version and machine-translated questions being the most striking: the accuracy increases from 21.0% to 34.4%. The accuracy of our simple baseline drops from 26.9% to 23.6%; DeepPavlov's drop in quality is more significant – from 28.6% accuracy on RuBQ 1.0 to 19.8% on QSQ. These results indirectly confirm that expanding the dataset is useful – it allows to get more reliable evaluation results. It is interesting to note that machine translated questions sent to the English version of QAnswer show a better result on QSQ than the original questions on the Russian-language QAnswer (English QAnswer outperforms its Russian counterpart on the whole dataset, but the

difference is less pronounced). Perhaps this is due to a more advanced English-language system, or to the fact that for shorter and more frequent questions from QSQ we get better machine translations. All systems are expectedly better at handling simple (1-hop) questions that make up the bulk of the dataset. Even the proposed simple baseline copes with simple questions on par with more sophisticated systems. However, the approach explicitly assumes the structure of the query and can only handle questions that correspond to a limited set of popular Wikidata properties. Both versions of QAnswer perform best on complex questions. DeepPavlov's mediocre performance on complex questions can be explained by the fact that the eight patterns that the system operates on are poorly represented among the limited number of RuBQ's complex questions. Few correct answers returned by a simple baseline to complex questions can be considered an artifact. English QAnswer with machine-translated questions achieved the best score on the answerable questions: 28.2% of correct answers. The result suggests that there is an ample room for improvements in multilingual KBQA methods. The performance of the systems on unanswerable questions sheds some light on their strategies. All the systems in the experiment seem to build a SPARQL query from a question without analyzing the local Wikidata graph or post-processing the returned results. Due to its cautious strategy, our simple baseline does not return an answer to most of the unanswerable questions. In contrast, QAnswer seems to be a recall-oriented system and returns an answer to almost all questions in both English and Russian versions.

### 5.2   MRC Baselines

*Models.* Multiligual BERT (mBERT) is a Transformer-based language model pre-trained on the top-100 languages from Wikipedia [5]. Fine-tuned BERT models demonstrated state-of-the-art performance in many downstream NLP tasks, including MRC. Interestingly, BERT-based models show competitive performance in zero-shot cross-lingual setting, when a model is trained on the English data and then applied to the data in another language. Artetxe et al. analyze this phenomenon on multilingual classification and question answering tasks [1]. For our experiments, we fine-tune $BERT_{BASE}$ Multilingual Cased model[19] on three training sets: English SQuAD [18], Russian SberQuAD [10], and a Russian subset of TyDi QA Gold Passage [4]. The number of training question-paragraph-answer triples is 87,599, 45,328, and 6,490, respectively.

*Evaluation.* MRC model returns a continuous span as an answer for a given paragraph and question. The answer is evaluated against gold answer spans provided by crowd workers. Traditionally, token-based F1 measure of the best match (in case of multiple gold answers) averaged over all questions is used as evaluation metrics. In our case we do not have explicitly marked answer spans within paragraphs, but have a list of correct answer's labels and aliases. Note that considering the way the MRC collection was created, the relevant paragraph

---

[19] https://huggingface.co/bert-base-multilingual-cased

**Table 4.** MRC baseline results: F1 based either on word overlap (Tokens and Lemmas), or best longest common subsequence (LCS) between gold and system answers.

| Training set | Tokens | Lemmas | LCS |
|---|---|---|---|
| SQuAD (En, 0-shot) | 0.54 | 0.70 | 0.76 |
| SberQuAD (Ru) | 0.48 | 0.62 | 0.70 |
| TyDi QA (Ru) | 0.51 | 0.67 | 0.73 |

must contain an answer in one form or another (see Section 4 for details). Since Russian is a highly inflectional language, the surface form of the answer in the paragraph may differ from a normalized form in the list. We experimented with three approaches to quantify the match between gold answers and model responses: token-based F1, lemmatized token-based F1, and character-based F1. Token-based metrics treat gold and system answers as bags of words, while the character-based metrics calculate the longest common subsequence (LCS) between the gold and the system answers. In case of the lemmatized version, both gold answers and system responses are processed with *mystem*. We consider the lemmatized token-based F1 as the most reliable metrics; however, its disadvantage is the overhead of lemmatization.

*Results.* We applied the models to all 3,638 pairs of questions and relevant paragraphs in the dataset (several relevant paragraphs can be associated with a question). Table 4 reports the performance of the three models. Note that the relative performance of the models is consistent across all metrics. Surprisingly, the model trained on English SQuAD scores the best, while the model trained on a small Russian collection TyDi QA is quite competitive. Although SberQuAD is seven times larger than Russian TyDi QA, it performs worse, probably due to a high level of noise in the annotations as we mentioned in Section 2. Although these scores do not account for the paragraph retrieval step and cannot be compared directly with KBQA scores in Table 3, we believe that hybrid KB/text approach to QA can substantially improve the overall results on the dataset.

## 6   Conclusions

In this work, we described the extension of RuBQ, the Russian dataset for question answering over Wikidata. The first version of the dataset was based on quiz questions and answer pairs harvested on the web. After exhausting this source of questions, we turned to search query suggestion services. This approach proved to be quite efficient: it required manual preparation of question prefixes and a later limited in-house verification; most of the annotation was carried out using crowdsourcing and automated routines. We managed to double the size of the dataset – from 1,500 to 2,910 questions. In addition to questions and Wikidata answers, the dataset contains SPARQL queries, tags indicating query type, English machine translations of the questions, entities in the question, answer aliases, etc. The dataset is accompanied by a Wikidata snapshot containing

approximately 8M entities and 212M triples, which ensures the reproducibility of the results. We evaluated three third-party and one our own KBQA systems on RuBQ. All systems are built on different principles and reflect well the range of approaches to KBQA. Based on the experimental results, we can conclude that the expanded dataset allows for a more reliable evaluation of KBQA systems.

We also expanded the dataset with machine reading comprehension capabilities: most questions were provided with Wikipedia paragraphs containing answers. Thus, the dataset can be used to evaluate machine reading comprehension, paragraph retrieval, and end-to-end open-domain question answering. The dataset can be also used for experiments in hybrid QA, where KBQA and text-based QA can enrich and complement each other [19]. We have implemented three simple MRC baselines that demonstrate the feasibility of this approach.

The main disadvantage of the dataset is a small number of complex questions. In the future, we plan to address this problem and explore different approaches to complex questions generation [20, 8].

The dataset is freely distributed under the CC-BY-4.0 license and will be of interest to a wide range of researchers from various fields – Semantic Web, Information Retrieval, and Natural Language Processing.

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
