# OpenReview forum: "RuBQ 2.0: An Innovated Russian Question Answering Dataset"
_eswc-conferences.org/ESWC/2021/Conference/Resources_Track — ESWC 2021 Resources_

### Official Review · AnonReviewer3 · 2021-01-11
**The paper describes the RuBQ 2.0 dataset, an extension of the RuBQ dataset for knowledge base question answering.**

**Rating:** 1
**Confidence:** 3

**Review:**

In general, the submission is easy to read and understand, with some effort from the reader. There are, however, at times, really long paragraphs (see for example paragraphs describing results). I suggest the authors to rewrite these and have a clear separation between results, for example. Have one paragraph per message / result or observation.


Open questions and comments
- It is not clear why the models used to evaluate the dataset are not mentioned in the related work, together with a comprehensive analysis - do they achieve state of the art? In the same direction, section 5 should provide the rationale behind choosing these baselines.
- How exactly were the initial questions crafted? What impact might this manual step have on the development of the dataset?
- It would be nice to link the crafted question prefixes examples with the exact properties in Wikidata.
- It is not clear whether the questions that imply numerical and date answers are also linked to the top properties in Wikidata. In either case, the exact procedure needs to be made clear.
- What does the semi-automatic filtering mean? (end of page 4)
- Since crowdsourcing is at the core of the dataset creation, I advise the authors to compute at least some inter-rater reliability metrics, mention whether any evaluation of the annotators, instances, etc. has been performed. Percentages are not the most useful numbers to understand how reliable annotations are.
- To make the paper self-contained, some more details should be given about the linker.
- What kind of manual correction was performed for questions' properties and entities?
- It is not clear from the text how the numbers in Table 1 are given.
- How many workers annotated the second crowdsourcing task, regarding machine reading comprehension? Also, similarly as above, what is the reliability of the annotations?
- The link with hybrid QA in section 4 needs more details.
- What are the implications for only choosing relevant paragraphs and not the exact span that constitutes the answer to the question? Are there any impediments for using the dataset in this case?
- Isn't it a very time-consuming effort to manually verify all paragraphs in the dataset? Could this step be done at least semi-automatically?
- Little details are provided with regard to training/testing of the models.
- It is very difficult to see the link between the numbers in table 3 and the percentages in the text. It would be much easier if these results should be reported consistently.



**Anonymity:**

Yes, I would like my review to remain anonymous.

**Strong Points:**

- knowledge base question answering dataset in Russian, containing 2910 questions and answers, along with SPARQL queries; the dataset could also be suitable for machine reading comprehension
- extension of existing resources, both in terms of the dataset and semantic web, NLP techniques, crowdsourcing

**Subreviewer:**

I submitted this review.

**Weak Points:**

- in its current form, the paper is, at times, not self-sufficient; while I agree that reusing already methods and resource is the right approach, some details should be included in the current submission (more details in the review section)
- the process of creating the dataset lacks structure - many manual steps that are not well explained
- lack of reflection with regard to the quality, suitability, replicability and reusability of the dataset

---

> ### Author Rebuttal · Authors · 2021-01-30
>
> We thank the reviewer for the comments. Responses to the questions raised are below.
>
> We will take into account suggestions regarding the paper organization and the presentation of the content. We will review the text and split long paragraphs, add information to make the paper self-contained, provide details about models used to evaluate the dataset and the whole testing process, elaborate on numbers in Table 1, and re-arrange Table 3. We will pay special attention to the structure of the paper and description of the whole annotation process, clearly stating in-house/crowdsourced/automatic steps and their impact on data quality.
>
> A short comment on the models used in evaluation. Most KBQA research prototypes work with English and Freebase as a target KB. QAnswer and DeepPavlov that we used in the evaluation are only two available KBQA systems for Russian working with Wikidata. In the case of MRC we just took the solution that became a de facto standard -- mBERT fine-tuned on a QA dataset (we used three datasets -- English SQuAD, as well as Russian SberQuAD and TyDi QA). BERT fine-tuned on SQuAD (single model) is #12 on SQuAD 1.1 leaderboard, which proves that the solution is still very competitive (it was posted on October 5, 2018).
>
> Generation of question prefixes was performed by the authors and started from the Wikidata property aliases. Then we generated variants of question prefixes corresponding to the property, trying to address diversity and coverage. Many prefixes for the same property are almost identical -- they differ only in verb tense/form. We will provide more examples of prefixes, properties, and corresponding questions returned by the query suggestion services. Prefixes for questions that imply numerical and date answers were generated separately, there is a small overlap with top200 properties; we will provide more details on this. Filtering of duplicate questions and queries without question words was performed automatically, while annotation of questions as good/bad for KBQA was performed manually by the authors. We concluded that it will be hard to formalize the latter task for crowdsourcing. Besides, this manual in-house filtering was quite fast.
>
> Each task (incl. paragraph annotation) was assigned to three crowd workers. We will elaborate on the design of the crowdsourced tasks and provide agreement statistics. Since we had answers (and their variants) for each question, we considered it unnecessary to annotate spans within paragraphs. This made the annotation faster and cheaper. Not all paragraphs, but only those containing the answer as a string (which doesn't guarantee they are sufficient to answer the question) were annotated. The paragraphs were verified by crowd workers, not the authors.
>
> We can state the RuBQ dataset steadily gains traction. DeepPavlov used RuBQ when preparing their new version of Russian KBQA model (see link in the paper). RuBQ was used in an ESWC2021 submission ‘Approaches to Port Question Answering Systems over Knowledge Graphs to new languages’, see https://openreview.net/forum?id=T2tTzsoxu-m . We also have an agreement with Mikhail Galkin (MILA Quebec & McGill University) that RuBQ 2.0 will be used in a MOOC on Knowledge Graphs (https://ods.ai/tracks/kgcourse2021). According to GitHub statistics, the RuBQ project page had 284 views (69 unique visitors) in the last two weeks.
>
> Apart from presenting the RuBQ 1.0 at ISWC2020, we presented RuBQ at two online seminars in 2020: 1) at the DeepPavlov seminar series (see recording https://www.youtube.com/watch?v=dYmEoSnJUIQ) and 2) at the series of Crowd Science Seminars by Toloka AI (see https://www.meetup.com/crowd-science-seminar/events/274818712/). We received very positive feedback. Current work improves RuBQ in terms of size and diversity, as well as adds MRC capabilities. We believe that rich and versatile annotation of the dataset will attract researchers from different communities -- Semantic Web, Information Retrieval, and Natural Language Processing.

---

> > ### Comment · AnonReviewer3 · 2021-02-03
> > **Answer to rebuttal**
> >
> > I appreciate the answers provided by the authors in this rebuttal. I would really like to encourage the authors to include all these changes and details in the paper to further strengthen it.

---

### Official Review · AnonReviewer1 · 2021-01-14

**Rating:** 1
**Confidence:** 4

**Review:**

This resource track paper presents a dataset to evaluate Russian QA systems constructed in a different way (i.e. leveraging commercial web search engine query suggestion functionalities) as compared to previously available datasets. Answers for questions are collected by means of crowdsourcing.

The process of constructing the dataset is well explained making the effort reproducible. The new dataset is compared in terms of descriptive statistics and experimental results with existing related ones.

In summary, the presented dataset appears to be a solid and useful contribution to the research community especially because it allows the comparison of results over multiple datasets.

Comments:
- it would be good to provide more details about the quality of the crowdsourced answers.

**Anonymity:**

Yes, I would like my review to remain anonymous.

**Strong Points:**

- solid methodology of constructing the dataset
- well presented method

**Subreviewer:**

I submitted this review.

**Weak Points:**

- unclear level of data quality

---

> ### Author Rebuttal · Authors · 2021-01-30
>
> We thank the reviewer for the comments.
> We will provide detailed statistics about crowdsourced annotation and resulted data quality.

---

### Official Review · AnonReviewer4 · 2021-01-15
**RuBQ review**

**Rating:** 2
**Confidence:** 4

**Review:**

The authors introduce an extension of their data set for answering Russian questions over knowledge graphs. They nearly double the size of the first version (1,410 new questions) by collecting questions from search engine suggestions and also add means for machine-reading comprehension tasks. The data set is of high quality and involved lots of manual effort during its creation. My main issue is with the evidence of usage of the data set, given that a first version does already exist.

Post-rebuttal comments:

* The authors provide a solid lists of examples where the first version of the dataset has actually been used. Therefore, I agree that it is of practical use and an extension seems reasonable.
* Thanks for the clarification of labels vs aliases in the data set.
* Generation of question prefixes: This still sounds rather fuzzy and hard to reproduce, but given that a large amount and variety of prefixes was created, the results seem fine at least.
* The author suggest few minor extensions (clarification of English/Russian prefixes + examples of search completion, clarification of label vs aliases, more statistics on the majority voting). These extensions should clarify most of my open questions.

Given the rebuttal and specifically the author's list of actual uses of RuBQ 1.0, I change my rating from weak accept to accept.

**Anonymity:**

Yes, I would like my review to remain anonymous.

**Strong Points:**

* Hybrid approach: Having one data set both for MRC and KGQA is interesting and useful.

* Data set: The data set in general is of high quality and contains nearly 3,000 manually verified Russian questions and answers. Also, the inclusion of a Wikidata sample helps evaluation against this data set in the future. nother interesting aspect is the availability of unanswerable questions.

* Effort: There was a lot of effort involved in creating this data set, including creation of query templates and crowdsourcing for relevant paragraph identification.


**Subreviewer:**

I submitted this review.

**Weak Points:**

* Motivation: The review guidelines explicitly ask for the evidence of usage or its potential. In this paper, the authors extend an existing dataset, so I was expecting information about how the first version was used. Related to this, it is not totally clear what was the motivation behind the extension. What can be done with RuBQ 2.0, but not with RuBQ 1.0? Have the authors used RuBQ 1.0 since its publication?

* Introduction/Method: Apart from the hybrid approach (MRC and KGQA), the biggest change towards RuBQ 1.0 is the use of query suggestion services. I would have liked to get more detail about this, as it is a core part of the approach. For example, a figure (e.g., a screenshot of a search engine) in the description that shows resulting queries given some prefix would help the reader's intuition. This gets particularly important at the beginning of Section 3, which is about question prefixes, although the example prefixes do actually contain the variable within the "prefix" ("What instrument did X play"). I assume this comes from the translation of Russian prefixes in the English, but clarification is needed.

* Data set: The machine translations of English queries are (as expected) not always perfect (specifically in the case of entity names, e.g., "Who created the site in Contact?" which asks about VKontakte). After there was so much manual effort and crowdsourcing involved in the process of creating the data set, I wonder why there was no correction of the English queries performed. I also think that separation between proper entity labels and aliases in the JSONs would help. In the GitHub description (and maybe as readme also in the zenodo dump), I would like to see a sample or explanation of an example. This was nicely done on the GitHub page of RuBQ 1.0, but not for the new version, which also contains new data to introduce (paragraphs).

* Creation of query prefixes: The creation of query prefixes is done manually but seems a bit arbitrary, and the process appears intransparent (e.g., "we compiled 546 question prefixes"). If I understand correctly (200 properties, 2077 prefixes), you collect more than 10 prefixes per property on average? Following what process?

* MRC: As pointed out at the strong points, the hybrid approach of proving knowledge graph answers and (ir)relevant paragraphs is interesting. However, these two aspects should be treated with the same attention. When reading the paper, I get the impression that MRC was added later on and does not perfectly fit into the reading flow. For example, the abstract starts with KBQA only, the data set statistics (Table 1) do not provide statistics about paragraphs, and the MRC part of the evaluation is a bit hard to follow.

* Data annotation: In Section 3, it is unclear how majority voting is applied in the case of multiple possible answers (do the whole lists of answers need to be identical?). In general, the statsitics in the "Answers" section could be presented in a table.

Minor:
* Title: "Innovative" instead of "Innovated"
* Abstract: "Th[is] paper"
* Introduction: "questions [posed] in a"
* Introduction: "from the version one"
* Section 2: "anew"
* Section 2: The description of MKQA could be slightly shortened.
* Section 3: The "target number" comes out of nowhere.
* Table 2: no need to show an example from version 1
* Table 2: make clear that these are paragraph identifiers
* Table 3: Explain in the caption what the numbers mean
* Section 5.1: "QAanswer"

---

> ### Author Rebuttal · Authors · 2021-01-30
>
> We thank the reviewers for their comments and questions. Responses to the questions raised are below.
>
> The motivation behind RuBQ 2.0 was larger size, higher diversity of questions, as well as the MRC part.  We believe that rich and versatile annotation of the dataset will attract researchers from different communities -- Semantic Web, Information Retrieval, and NLP.
>
> We have been advertising RuBQ during the second half of 2020 though it was severely hindered by the covid-19 pandemic. Apart from presenting the RuBQ 1.0 at ISWC2020, we presented the dataset at two online seminar series: 1) DeepPavlov seminar (see recording https://www.youtube.com/watch?v=dYmEoSnJUIQ) and 2) Crowd Science by Toloka AI (see https://www.meetup.com/crowd-science-seminar/events/274818712/). DeepPavlov used RuBQ when preparing their new version of Russian KBQA model (see link in the paper). RuBQ was used in an ESWC2021 submission ‘Approaches to Port Question Answering Systems over Knowledge Graphs to new languages’, see https://openreview.net/forum?id=T2tTzsoxu-m . We also have an agreement with Mikhail Galkin (MILA Quebec & McGill University) that RuBQ 2.0 will be used in an upcoming MOOC on Knowledge Graphs (https://ods.ai/tracks/kgcourse2021). According to GitHub statistics, the RuBQ project page had 284 views (69 unique visitors) in the last two weeks. So we can state the RuBQ dataset steadily gains traction.
>
> We will provide examples of SE responses for several prefixes (as we use an API, it’s not possible to present them as a screenshot). Indeed, the prefixes don’t contain variables -- the Xs appear only in English translations, we will elaborate on this.
>
> According to our past experience, manual post-editing of machine translation can be quite laborious, requires skilled crowd workers, and additional budget, so we abandoned this idea.
>
> Separation between proper entity labels and aliases is provided implicitly. The ‘label’ field in JSON stays for the actual label, while ‘wd_names’ contain all the aliases + the label. So obtaining only aliases can be done by removing the label token from wd_names list. We will make a note about it in the format description and provide an example of a dataset entry like we did it for RuBQ 1.0.
>
> Generation of question prefixes was performed by the authors and started from the property aliases from Wikidata. Then we generated variants of question prefixes corresponding to the property, trying to achieve diversity and coverage. Many prefixes for the same property are almost identical -- they differ only in verb tense/form.
>
> In case of multiple answers majority voting was applied to individual answers, not the whole list. We will arrange statistics from the Answers section in a table.
>
> We are grateful for suggested corrections and will account for them in the next version of the paper.

---

### Decision · Program_Chairs · 2021-02-23

**Decision:**

Accept

**Comment:**

The paper introduces a second version of the RuBQ dataset, a Russian dataset for knowledge base question answering. All reviewers suggest that the paper should be accepted, although they suggest that notes from the authors' rebuttal should be included in the final version.  Reviewers raised issues regarding the need for a new version, issues with the paper not being self-contained, issues with the quality of some of the data, e.g., the English translations, and so on. On the other hand, they praise the effort in creating the dataset, the size of the dataset, and the methods used to create it.